# Differential Kinetics of Effector and Memory Responses Induced by Three Doses of SARS-CoV-2 mRNA Vaccine in a Cohort of Healthcare Workers

**DOI:** 10.3390/vaccines10111809

**Published:** 2022-10-27

**Authors:** Federica Bergami, Francesca Arena, Josè Camilla Sammartino, Alessandro Ferrari, Federica Zavaglio, Paola Zelini, Stefania Paolucci, Giuditta Comolli, Elena Percivalle, Daniele Lilleri, Irene Cassaniti, Fausto Baldanti

**Affiliations:** 1Microbiology and Virology Department, Fondazione IRCCS Policlinico San Matteo, 27100 Pavia, Italy; 2Experimental Research Laboratories, Biotechnology Area, Fondazione IRCCS Policlinico San Matteo, 27100 Pavia, Italy; 3Department of Clinical, Surgical, Diagnostics and Pediatric Sciences, University of Pavia, 27100 Pavia, Italy

**Keywords:** SARS-CoV-2 vaccination, immune response, healthcare workers

## Abstract

We reported the long-term kinetics of immune response after vaccination and evaluated the immunogenicity after a third dose of mRNA vaccine in 86 healthcare workers. Humoral response was analyzed by measuring anti-spike IgG and SARS-CoV-2 NTAbs titer; cell-mediated response was measured as frequency of IFN-γ producing T-cells and cell proliferation. Memory B cells secreting SARS-CoV-2 RBD-IgG were measured by B-spot assay. At three weeks after the third dose (T4), the frequency of subjects showing NT-Abs titer at the upper detection limit (≥640) was significantly higher than that observed at three weeks after the second dose (26/77; 33.7% vs. 9/77; 11.6%; *p* = 0.0018). Additionally, at T4, all the subjects reached positive levels of T-cell mediated response (median 110 SFU/10^6^ PBMC, IQR 73-231). While the number of IFNγ-producing T-cells decreased between second and third dose administration, the T-cell proliferative response did not decrease but was sustained during the follow-up. Among T-cell subsets, a higher proliferative response was observed in CD4+ than in CD8+ population. Moreover, even if a decline in antibody response was observed between the second and third dose, a sustained persistence of memory B cells was observed. Subsequently, the third dose did not affect the frequency of memory B cells, while it restored or increased the peak antibody levels detected after the second dose.

## 1. Introduction

Severe acute respiratory syndrome Coronavirus 2 (SARS-CoV-2) still represents a crucial health problem, despite the introduction and the massive use of vaccines. Since SARS-CoV-2 was identified [1], the coronavirus disease 2019 (COVID-19) has been detected in more than 476 million cases in the world, causing more than 6 million related deaths. So far, vaccination is the only effective option to control the spreading of COVID-19.

The mRNA BNT162b2 vaccine [2] showed 95% protection against a SARS-CoV-2 infection in a phase II/III trial [3]. Then, another mRNA-based vaccine, mRNA-1273 [4], showed similar results. Several studies reported a sustained humoral and cell-mediated response elicited by a two-dose schedule [5,6,7,8,9]. However, the emergence of SARS-CoV-2 variants of concern (VOCs) has raised concerns about the immunogenicity of the vaccine and about the persistence of the sustained immune response [10].

In order to sustain an effective immune response and control further epidemic waves, a booster dose was administered, starting from immunocompromised patients and healthcare workers, resulting in a decrease in hospitalizations and deaths. A number of studies showed that the rate of confirmed infection was lower in subjects who received a booster of vaccine than in control subjects [11,12,13].

So far, the administration of a booster dose seems to also be effective in increasing immunogenicity against the SARS-CoV-2 VOCs Delta [14] and Omicron [15]. In this setting, Wu et al. have demonstrated that after the third dose, 96% of subjects had detectable neutralizing antibodies (NT Abs) against Omicron, while before the booster only 42% did [15].

In this paper, we reported the long-term kinetics of humoral and cell-mediated response after vaccination and evaluated the immune response after the booster dose in the same healthcare workers.

## 2. Materials and Methods

### 2.1. Study Population

We designed a prospective study to evaluate the immune response elicited by a full course of mRNA vaccination (BNT162b2 vaccine) against SARS-CoV-2, including a third booster dose, in 86 healthcare workers (68 female, 18 male), median age 48.5, range 25–69 years of age. In this group, 77/86 (89.5%) subjects were defined as “SARS-CoV-2- naïve” since they did not experience a SARS-CoV-2 infection before the vaccination (anti-N IgG and/or anti-Spike IgG-negative) while 9/86 (10.5%) were defined as SARS-CoV-2-experienced since they recovered from a SARS-CoV-2 infection before vaccine administration (anti-N IgG and/or anti-Spike IgG-positive).

Samples were collected at the time of the first mRNA vaccine dose between December 2020 and January 2021. The follow-up time points were defined as follows:(i)T0 (baseline, the day of first dose administration);(ii)T2 (42 days after first dose; median day 41, range 39–53 days);(iii)T3 (six months after first dose; median day 187 range 148–239 days);(iv)T4 (21 days after third dose administered approximatively 9 months after the first dose; median day 287 range 268–338 days).

The study (CoVax) was approved by the local Ethics Committee (Comitato Etico Area Pavia) and Institutional Review Board (P-20210000232). All the subjects signed informed written consent.

### 2.2. Evaluation of Spike-Specific IgG and Neutralizing Antibodies

Serum was used for Spike SARS-CoV-2 IgG serology and neutralizing antibodies (NT-Abs) titer. SARS-CoV-2 IgG levels were measured using chemiluminescence immunoassay (Diasorin, Saluggia, Italy, SARS-CoV-2 Trimeric S IgG). The sample was considered positive when the IgG level was higher or equal to 33.8 BAU/mL. At T4, N-SARS-CoV-2 total IgG levels were measured using chemiluminescence immunoassay (Roche Diagnostics, Rotkreuz, Switzerland). The sample was considered positive when ≥1.2 U/mL. SARS-CoV-2 NT-Abs were evaluated through microneutralization assay as previously described [16] against the reference strain (hCoV-19/Italy/LOMINMI-10734/2020) and the Omicron strain (hCoV-19/Italy/LOMPavia-10943/2021)), previously isolated and titrated. Briefly, 50 µL of sample from each patient, starting from 1:10 in a serial fourfold dilution series, was added in two wells of a flat bottom tissue culture microtiter plate (COSTAR, Corning Incorporated, Corning, NY, USA), mixed with an equal volume of 50 TCID50 of a SARS-CoV-2 strain isolated from a symptomatic patient, previously titrated and incubated at 33 °C in 5% CO_2_. All dilutions were made in EMEM with addition of 1% penicillin, streptomycin and glutammin and 5 γ/mL of trypsin. After 1 h incubation at 33 °C 5% CO_2_, VERO E6 cells (VERO C1008 (Vero 76, clone E6, Vero E6); ATCC^®^ CRL-1586™) were added to each well. At 48 h of incubation at 33 °C and 5% CO_2_, the wells were stained with Gram’s crystal violet solution (Merck KGaA, Damstadt, Germany) plus 5% formaldehyde 40% m/v (Carlo ErbaSpA, Arese, Italy) for 30 min. Microtiter plates were then washed in running water. Wells were scored to evaluate the degree of cytopathic effect (CPE) compared to the virus control. NT Abs titer was defined as the maximum dilution with the reduction of 90% of CPE. A positive titer was equal to or greater than 1:10.

### 2.3. Spike-Specific T and B Cell Responses

Peripheral blood mononuclear cells (PBMCs) were isolated from heparin-treated blood by standard density gradient centrifugation and used for ELISpot assays. Spike-specific T-cell response was evaluated using ex vivo ELISpot assay after culture with Spike (S)-specific peptide pool [17]. In detail, membrane-bottomed 96-well plates (Multiscreen-IP) from Merck Millipore, Germany, were coated with anti-interferon (IFN)-γ monoclonal capture antibody from Human IFN-γ ELISpot kits (Diaclone, France) and kept at 4 °C overnight. Then, after 2 hr blocking with culture medium, 200,000 cells/100  μL per well were stimulated with antigens; phytohemagglutinin (PHA, 5 μg/mL, Sigma-Aldrich) and medium alone were used as positive and negative control, respectively. All the experiments were performed in duplicate. Plates were maintained overnight at 37 °C in a 5% CO_2_ humidified atmosphere. After multiple washes, anti-IFN–γ biotinylated antibody was added and incubated overnight at 4 °C. Finally, streptavidin–alkaline phosphatase conjugate was added, and after 60 min incubation at 37 °C in a 5% CO_2_, substrate 5-bromo-4-chloro-3-indolyl phosphate/nitro blue tetrazolium (BCIP/NBT) was added for 20 min at room temperature. Plates were washed under running water and kept overnight at room temperature before spot counting. AID ELISPOT reader system from Autoimmun Diagnostika GmbH (Strasburg, Germany) was used for count. Results equal to or higher than 10 IFN-γ SFU/10^6^ PBMC were considered positive.

Spike-specific proliferative response was determined as described below. Briefly, PBMCs (600,000/200 μL culture medium per well) were stimulated in triplicate with S and human actin peptide pools (15 mers, overlapping by 10 amino acids, Pepscan, Lelystad, The Netherlands) at a final concentration of 0.1 µg/mL for 7 days. After culture, cells were washed, stained with Live/Dead Fixable Violet Dye (Invitrogen, Waltham, MA, USA) and subsequently with CD3 PerCP 5.5 (BD Bioscience, Franklin Lakes, NJ, USA), CD4 APC-Cy7, CD8 FITC, CD25 PECy7 (all from BD Bioscience), and CD278 (ICOS) APC (Invitrogen). Finally, cells were washed and resuspended in PBS 1% paraformaldehyde. A cell proliferation index (CPI) was determined by subtracting the percentage of CD25 + ICOS + CD3 + CD4+ or CD3 + CD8+ detected in PBMC incubated with actin peptides from the percentage of CD25 + ICOS+ T-cell subsets detected in PBMC incubated with S peptides. A CPI ≥1.5% was considered positive [18]. Flow cytometry analyses were performed with a FACS Canto II flow cytometer and BD DIVA software (BD Biosciences).

For the enumeration of memory B cells secreting IgG antibodies specific to the SARS-CoV-2 receptor binding domain (RBD) and the enumeration of all B cells secreting IgG (total IgG), a commercial kit was used according to manufacturers’ instructions (MABTECH, ELISpot Path: Human IgG ALP). Results were given as numbers of RBD specific spots/number of total IgG spots and values higher than 2.85 were considered positive.

### 2.4. Routine Surveillance for SARS-CoV-2 RNA Detection

Naso-pharyngeal swabs were collected and tested for SARS-CoV-2 RNA positivity in subjects with symptoms suggestive of SARS-CoV-2 infection or in case of contact with infected subjects. SARS-CoV-2 RNA real time PCR was performed weekly or every two weeks, using SARS-CoV-2 ELITe MGB**^®^** Kit (ELITechGroup, Puteaux, France). Moreover, in compliance with the local healthcare workers’ surveillance protocol, personnel working in clinical wards dedicated to fragile patients underwent a routine screening for SARS-CoV-2 infection every 14 days, while monitoring was scheduled every 30 days for healthcare workers in the other wards. The health condition of all workers was regularly monitored and data on symptoms were collected during an interview by a physician and inserted into a specific database.

### 2.5. Statistical Analyses

Median, range and interquartile range were reported for continuous data while frequency and percentages were used for qualitative data. The comparison of levels of SARS-CoV-2-specific immunological parameters detected at each time point was performed with the Friedman test for repeated measures (with correction for multiple comparisons and Dunn’s post-test). Fisher’s exact test was adopted for the comparison of frequencies. All analyses were performed using Prism 8.3.0 (GraphPad software, San Diego, CA, USA).

## 3. Results

### 3.1. B Cell Response in SARS-CoV2-Naïve and -Experienced Vaccinated Subjects

At T2, all SARS-CoV-2-naïve subjects developed positive levels of anti-spike antibodies, and 42/77 (54.5%) of them showed levels at the upper limit of the quantifiable range of the assay (>2080 BAU/mL). At T3, positive IgG response was still observed in all the subjects, although the levels significantly decreased (median level: 424.4, IQR 278.2–686.4 BAU/mL; *p* < 0.0001) and all the subjects were within the quantifiable range. At T4, after the booster dose, the levels increased again and the frequency of subjects showing levels >2080 BAU/mL (72/77; 93.5%) was significantly higher than that observed at T2 (*p* < 0.0001) (Figure 1A,B). The NT Abs titer followed the same kinetics. At T2, all naïve subjects showed a positive response of NT Abs and 9/77 (11.6%) subjects showed levels above the assay range (≥640). NT Abs titer decreased at T3, when all but two subjects still showed positive values (median titer: 40 IQR 20–80; *p* < 0.0001). Finally, at T4, the NT Abs titers increased again, returning positive in all the subjects (Figure 1C,D). All the nine SARS-CoV-2-experienced subjects showed anti-Spike antibodies >2080 BAU/mL at T2. Levels decreased at T3, when 3/9 (33.3%) subjects were >2080 BAU/mL (*p* = 0.0090), and increased again at T4, when 8/9 (88.9%) subjects were >2080 BAU/mL.

The statistical analysis of quantitative levels was not possible because almost all subjects at T2 and T4 were above the quantifiable range. At T4, the frequency of subjects showing NT Abs titer at the upper detection limit (≥640) was significantly higher than that observed at T2 (26/77; 33.7% vs. 9/77; 11.6%; *p* = 0.0018). In SARS-CoV-2-experienced subjects, NT Abs positive values were detected at all follow-ups. Looking at NT Abs titer against the Omicron variant, a positive NT Abs level was observed in 96% of the subjects after third-dose administration (T4), with a median NT Abs titer of 40 (IQR 20–160) (Figure 2).

The same trend was observed in SARS-CoV-2-experienced subjects. All the nine SARS-CoV-2-experienced subjects showed NT Abs titer ≥640 at T2 and T4, whereas in 3/9 (33.3%) subjects the NT Abs titers were ≥640 (*p* = 0.0090). As expected, the median levels of anti-Spike IgG and NT Abs were significantly higher in SARS-CoV-2-experienced subjects than in naïve subjects at T3 but not at T4. The frequency of Spike-specific memory B cells secreting IgG antibodies was evaluated in 25 subjects at T2 and T3 and in 8 subjects at T4 (due to the low number of cells available). Conversely to what was observed for the antibody levels, the memory B cells did not decrease before booster administration, but a progressive increase in the proportion of subjects with detectable Spike-specific memory-B cells was observed (chi-squared test for trend: *p* = 0.036). At T2, 17/25 (68%) subjects showed detectable Spike-specific memory B cells (median frequency 3.6; IQR 1.2–11.5), which was the case at T3 for 21/25 (84%) subjects (median frequency 10.36; IQR 4.1–21; *p* = 0.0847 vs. T2) and for the remaining eight subjects at T4 (median frequency 28.2, IQR 9.9–32; *p* = 0.0074 vs. T2) (Figure 3).

### 3.2. T-Cell Response in SARS-CoV-2-Naïve and -Experienced Vaccinated Subjects

Spike specific T-cell response was evaluated in 82/86 subjects (95.3%; 74 SARS-CoV-2 naïve and 8 SARS-CoV-2 experienced). At T2, in all but one SARS-CoV-2-naïve subject T-cell response was positive (median frequency: 110 SFU/10^6^ PBMC, IQR 49–175) and the large majority of them (67/74; 90.5%) maintained a positive Spike T-cell response at T3, although the levels decreased (median frequency: 30 SFU/10^6^ PBMC, IQR 15–75; *p* < 0.0001). After the booster dose (T4), the frequency of Spike-specific T-cell increased again to levels similar to T2 (median frequency: 110 SFU/10^6^ PBMC, IQR 73–231 (Figure 4A)). All SARS-CoV-2-experienced subjects preserved a Spike-specific T-cell response during all follow-ups and no significant differences were detected. In particular, no decline was observed at T3 (Figure 4B). While the number of IFNγ-producing T-cells decreased at T3, before increasing again after the booster dose, the T-cell proliferative response did not decrease but was sustained during the whole follow-up, since no significant differences were observed across the time points (Figure 4C,D).

### 3.3. SARS-CoV-2 Infection in Vaccinated Subjects

SARS-CoV-2 was not detected in the nasal swabs of any subject between T2 and T4. In addition, at T4, all SARS-CoV-2-naïve subjects showed a negative value for N-SARS-CoV-2 total IgG, suggesting that no SARS-CoV-2 infection occurred during the period of follow-up, during which the Apha and Delta VOCs were circulating. Subsequently, 26 subjects developed a SARS-CoV-2 infection (diagnosed by nasal swab testing) in a median time of 80 days (range 61–117) after the booster dose, during the circulation of the Omicron VOC. The infection was mildly symptomatic in all subjects, none of whom required hospitalization or antiviral treatment. Levels of antibody and T-cell response after the booster dose in infected or non-infected subjects are shown in Figure 5A–C.

## 4. Discussion

In summary, the results of this study show that the antibody levels decreased six months after the administration of the second dose and increased again after the booster dose, reaching higher levels than those observed after the second dose. Conversely, the frequency of memory B cells able to secrete RBD-specific IgG antibodies after in vitro re-stimulation did not decline, but progressively increased. Similarly, while Spike-specific T-cells with a rapid effector function (i.e., IFNγ production) decreased six months after the second dose, Spike-specific T-cell proliferation (a hallmark of memory cells) remained stable throughout the follow-up, with higher response in the CD4+ than in the CD8+ compartment. Similarly, in an 8-month follow-up, the peak of humoral and cell-mediated response after the first vaccine dose was observed at 29 days, with a progressive decline during the follow-up period [19].

Additionally, even after the booster dose administration, the overall NT Abs titer elicited against the Omicron variant remained lower, as previously demonstrated [20,21]. The decline in antibody levels a few months after vaccination has been consistently reported [22,23]. Nevertheless, our data, in agreement with other studies [24,25] show a sustained persistence of memory B cells 6 months after the second dose, which is dissociated from the decline in circulating antibody levels. Subsequently, the third dose did not affect the frequency of memory B cells, while it restored (or even increased) the peak antibody levels detected after the second dose. Similarly, it was observed that in SARS-CoV-2-experienced subjects, vaccine administration boosted the pre-existing immunity inducing an increase in antibody levels with no or little change in memory B cell frequency [25]. This observation suggests that the third dose did not, or did so minimally, elicit the generation of new memory B cells from the naïve pool, but rather it induced a rapid recall response from pre-existing memory B cells. Whether this will be followed by a more sustained antibody production by long-living plasma cells should be verified in future studies. However, the observation that antibody levels persist at higher levels in SARS-CoV-2-experienced than in naïve subjects endorses this hypothesis. The persistence of memory B cells induced by a mRNA vaccine appears more sustained than that observed after the administration of an inactivated SARS-CoV-2 vaccine [26,27], in which case memory B cells are still detectable at six months but their frequency decreases. It is tempting to speculate that immune memory induced by mRNA vaccines is similar to that induced by live-attenuated vaccines, which is longer-lasting than that induced by inactivated or subunit vaccines. We can exclude that memory B cells were maintained due to an incurrent subclinical SARS-CoV-2 infection since all subjects tested negative for anti-N antibodies before the administration of the booster dose.

Among T-cell subsets, a higher proliferative response was observed in the CD4+ than in the CD8+ population. Some authors hypothesized that low CD8+ T-cell activation might be due to an inadequate Th1 response than respective to a Th2 response [28]. Interestingly, a sustained proliferative response was observed in both the naïve and SARS-CoV-2-experienced subjects. Understanding the long-term persistence of the memory response elicited by mRNA-based vaccines is an important issue, since this is a novel platform which has been widely used for the first time for SARS-CoV-2 vaccination, but its use is likely to expand in the next future, as many trials are ongoing for infectious or non-infectious disease [29,30]. The kinetics of the T-cell is similar to that of the B cell response. The frequency of IFNγ-producing effector T-cells decreases with time but is boosted by the third dose. We measured T-cell proliferation as a surrogate marker of memory T-cells, and this activity did not decrease with time, nor was it increased by the booster dose. We did not measure the actual frequency of memory T-cells able to proliferate after recall stimulation, but although it is possible that the number of memory T-cell declines while their proliferative potential increases, it is more likely that a steady memory T-cell pool is maintained.

The declining antibody and effector T-cell levels may with time be the basis for breakthrough infections following vaccination. However, in this as well as in a previous study [31], we observed that protection from a breakthrough SARS-CoV-2 infection is not associated with the quantity of anti-Spike antibodies or T-cells. In addition, the persistence of memory B cells suggests that future heterologous booster doses with vaccine formulation based on new Spike variants may be able to stimulate cross-reactive B cells (or B cells recognizing conserved epitopes) and induce a broader anti-SARS-CoV-2 activity. The persistence of memory T-cells is even more important for the rapid control of new variants, since these cells are less affected by the mutations in the Spike that reduces antibody efficacy.

The limitations of this study reside in the low number of SARS-CoV-2-experienced subjects analyzed. In addition, we did not study the immune response kinetics in elderly or immunocompromised individuals. Regarding the total IgG and SARS-CoV-2 NT Abs, since the majority of the results reached the upper limit of assay detection at T2 and T4, quantitative analyses were difficult to interpret. On the other hand, the frequency of subjects who reached the upper limit of assay detection was considered a qualitative result. Looking at the B cell memory response, we were able to analyze only partial data, due to the lack of samples and the fact that no phenotypical data were provided in terms of T-cell memory response. Finally, the sequencing of the SARS-CoV-2 type in infected subjects after a third dose was not performed. However, considering the epidemiological data of SARS-CoV-2 VOCs circulation between January and March 2022, it is conceivable that the Omicron variant was responsible for the infections detected in our cohort of subjects [32]. Future studies should address the long-term persistence of memory response after the three vaccine doses, as well as the effect of additional booster doses with vaccines designed for the new SARS-CoV-2 variants.

## Figures and Tables

**Figure 1 vaccines-10-01809-f001:**
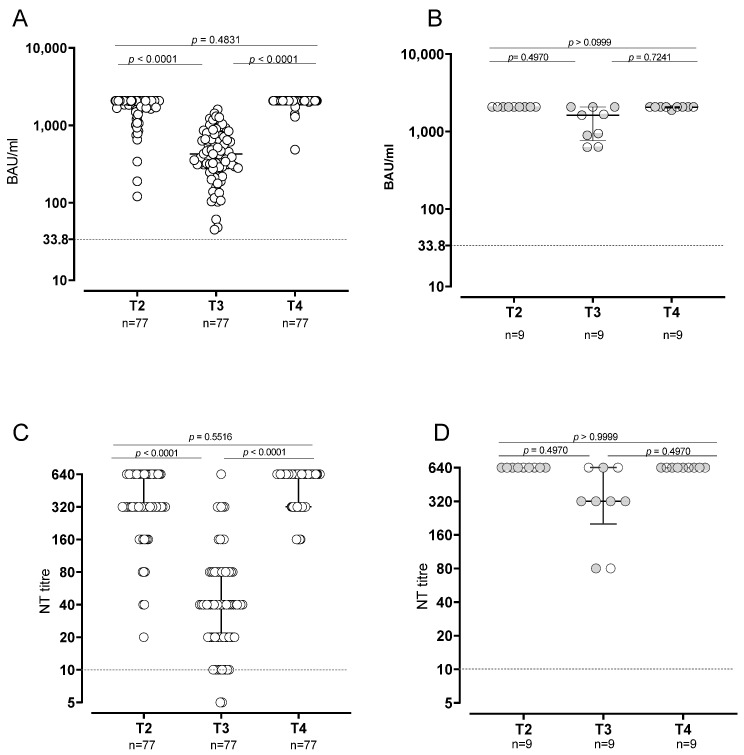
Humoral response in SARS-CoV2-naïve and -experienced vaccinated subjects. Levels of anti-spike antibodies in SARS-CoV2-naïve (**A**) and -experienced (**B**). SARS-CoV-2 neutralizing antibodies in -naïve (**C**) and -experienced (**D**) subjects.

**Figure 2 vaccines-10-01809-f002:**
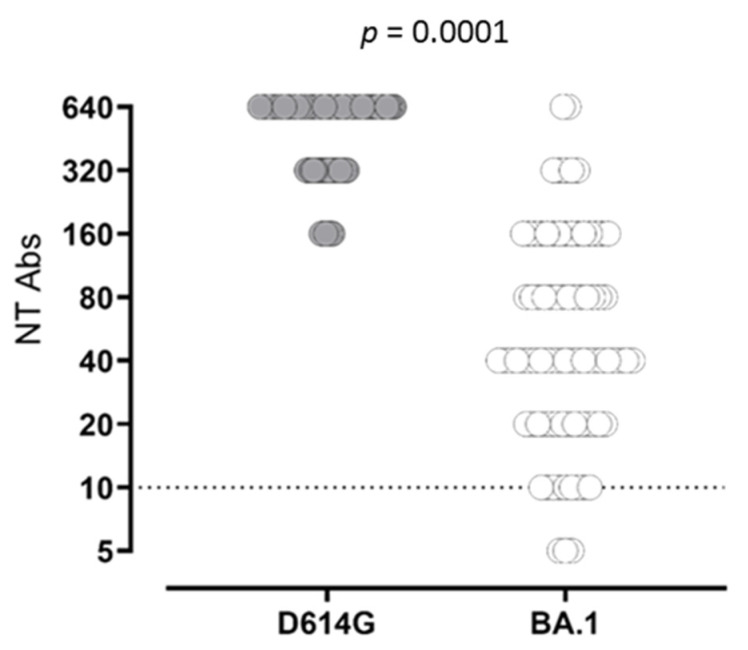
SARS-CoV-2 NT Abs response against wild-type strain D614G (grey dots) and BA.1 variant (white dots) was measured.

**Figure 3 vaccines-10-01809-f003:**
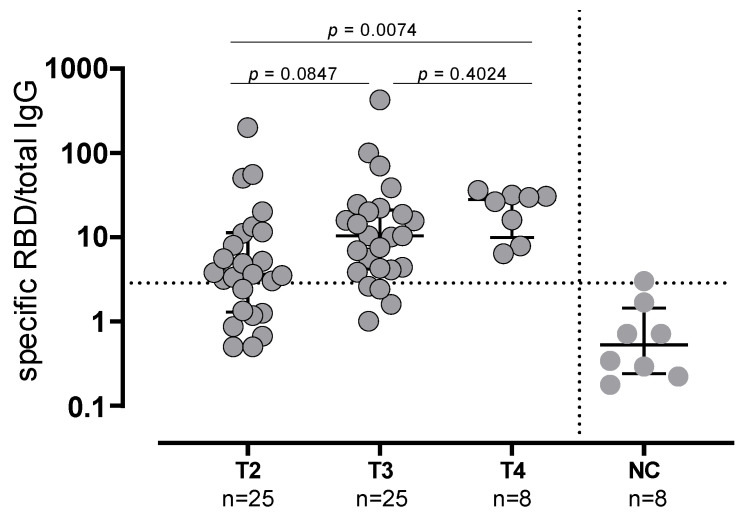
Spike-specific memory B cells secreting IgG antibodies evaluated in 25 subjects at T2 and T3 and in 8 subjects at T4. NC: negative controls.

**Figure 4 vaccines-10-01809-f004:**
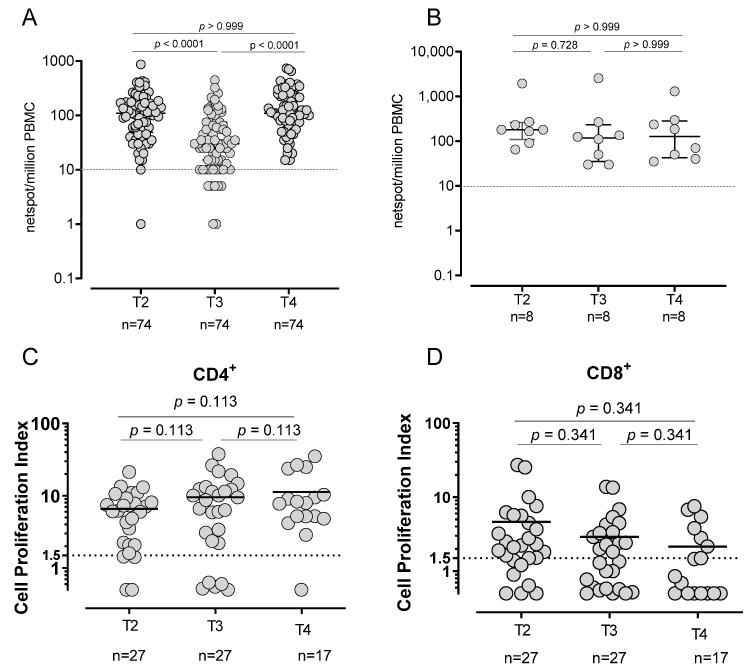
Spike-specific T-cell response. T-cell response in SARS-CoV-2-naïve (**A**) and -experienced (**B**) vaccinated subjects using Elispot assay. Spike-specific proliferative response in CD4+ (**C**) and CD8+ (**D**).

**Figure 5 vaccines-10-01809-f005:**
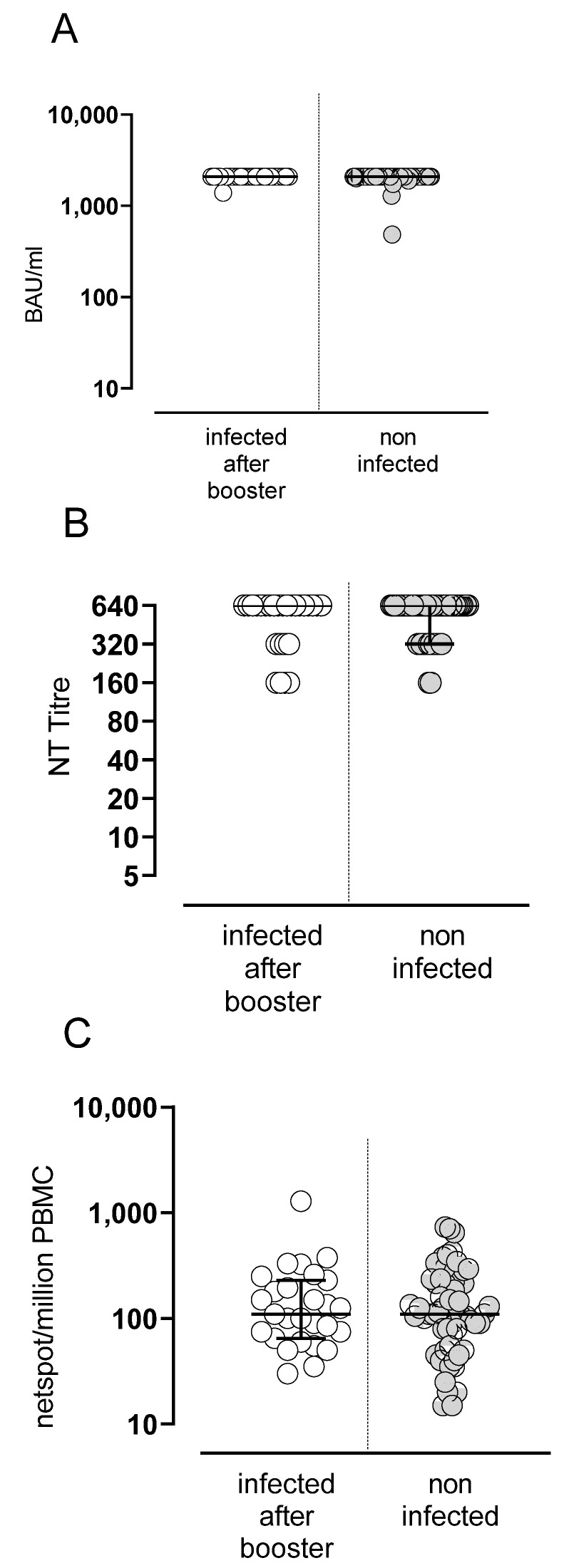
Spike-specific B and T-cell responses in SARS-CoV2 infection in vaccinated subjects. Levels of anti-spike IgG antibodies (**A**), NT Abs (**B**) and Spike-specific T-cell response (**C**).

## Data Availability

The data presented in this study are available on request from the corresponding author.

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
