# Peer review of "Differential Kinetics of Effector and Memory Responses Induced by Three Doses of SARS-CoV-2 mRNA Vaccine in a Cohort of Healthcare Workers"

_vaccines, 2022, doi:10.3390/vaccines10111809_

Round 1

Reviewer 1 Report

The paper by Bergami F. et al. entitled “Differential kinetics of effector and memory responses induced by three doses of SARS-CoV-2 mRNA vaccine in a cohort of healthcare workers” describes the results of a study aiming at evaluating the humoral and cell-mediated immune response after the administration of SARS-CoV-2 mRNA vaccine in 86 healthcare workers. The assessment was carried out on samples collected before the vaccination (T0) and after the administration of each vaccine dose (three time-point). The study population is a subgroup of a larger population considered in other studies of the same research group.

These are some comments for authors’consideration:

-          Abstract should be revised to be more informative and less narrative, it would be appreciated to read some quantitative results (i.e. the n. of study participants for example, the percentage of response)

-          The introduction should be completed with current knowledge on immune response to vaccination and adding the state of the art on SARS-CoV-2 mRNA vaccines.

-          The “Study population” should be described in more details. I’d suggest to add a figure or a table to have a clear representation of the number of samples available for testing at each time point. The brand of mRNA vaccine administered to study participants should be indicated: did all HCWs receive the same vaccine for the three doses? Also, the timing of the three doses administration should be reported. The time points should be clarified better and should be referred to the same baseline. It’s unlikely that all samples were collected on the exact day, a range of days should be included. for T0: specify if samples were collected before or after the vaccine administration.

-          in the material and methods section I‘d suggest to provide a brief description of the methods rather than referring to previous work. Also, the molecular detection of SARS-CoV-2 in the framework of active and passive surveillance of HCWs should be described in more details rather than referring to a general “local protocol”. Finally, the accession numbers of the sequences of the virus seeds used in MN assays should be included.

-          The reference reported should be carefully checked; reference no. 18 is reported in the main text but not in the reference list.

-          English language should be revised.

These are some minor comments:

Lines 31 and 34: the name in full of COVID-19 is reported twice

L. 41: When refer to VOC please add SARS-CoV-2 variant of concern

L. 41: Wu et al “have” demonstrated

L.44: we “have” reported

L.44: cell-mediated

L.45: SARS-CoV-2, revise it throughout the manuscript

L.54: add a reference to “our previous work”

L. 57: this sentence should be rephrased

L.70: use the acronym for neutralizing antibodies (revise it throughout the manuscript)

l. 71: use “described” rather than “reported”

L. 77: clarify how results were expressed

L. 82: specify “after culture”

L. 95: delate the bracket after spots

L.100: add references to protocols used for molecular detection of SARS-CoV-2

L.113. final dot is missing

L. 115: “vaccinated” rather than “vaccinate” (check it throughout the manuscript)

L.126: “at” rather than “At”

l. 137-138: this sentence should be clarified

L. 141-142: this sentence should be clarified

L. 152: the caption of figure 2 should be revised

L.168: add the meaning of “NC” in the caption of figure 3

L. 171: “subject” rather than “subjects”

L-176: SARS rather than Sars

L.180: this sentence should be clarified and report some quantitative data

L.182-185: this sentence should be moved to the discussion section. It is not relevant to the results section

L.190-193: this sentence should be moved to the discussion section. It is not relevant to the results section

L.203: “a negative value” rather than “a negative values”

L.205: refer to “Alpha” and “Delta” VOCs

L.209: “was similar” rather than “was not significantly different”

L.216: “the results” rather than “the major results”

L.220: in vitro re-stimulation

L.226: “has been reported” rather than “have been reported”

L.227: “studies” rather than “studies”

L.252: Reference no 18 is missing

L.255: delete brackets for SARS

L. 259: “limitations” rather than “limitation”

L.260: do you mean elderly?

L.275-277: missing

Author Response

The paper by Bergami F. et al. entitled “Differential kinetics of effector and memory responses induced by three doses of SARS-CoV-2 mRNA vaccine in a cohort of healthcare workers” describes the results of a study aiming at evaluating the humoral and cell-mediated immune response after the administration of SARS-CoV-2 mRNA vaccine in 86 healthcare workers. The assessment was carried out on samples collected before the vaccination (T0) and after the administration of each vaccine dose (three time-point). The study population is a subgroup of a larger population considered in other studies of the same research group.

These are some comments for authors’ consideration:

-          Abstract should be revised to be more informative and less narrative, it would be appreciated to read some quantitative results (i.e. the n. of study participants for example, the percentage of response)

Response: As suggested by the reviewer, the abstract section was revised including more quantitative data.

-        The introduction should be completed with current knowledge on immune response to vaccination and adding the state of the art on SARS-CoV-2 mRNA vaccines.

Response: We included more references in the introduction, as suggested                                                                                                                                                                                                                                                                               

-          The “Study population” should be described in more details. I’d suggest to add a figure or a table to have a clear representation of the number of samples available for testing at each time point. The brand of mRNA vaccine administered to study participants should be indicated: did all HCWs receive the same vaccine for the three doses?

Response: We included the number of samples analyzed for each test and for each time point in all the figures. Additionally, all the subjects included in the present study received only mRNA BNT162b2 vaccine, as specified in material and methods section.

  • Also, the timing of the three doses administration should be reported. The time points should be clarified better and should be referred to the same baseline. It’s unlikely that all samples were collected on the exact day, a range of days should be included. for T0: specify if samples were collected before or after the vaccine administration.

Response: Timing of samples for each time point was better described as requested.

-          in the material and methods section I‘d suggest to provide a brief description of the methods rather than referring to previous work. Also, the molecular detection of SARS-CoV-2 in the framework of active and passive surveillance of HCWs should be described in more details rather than referring to a general “local protocol”. Finally, the accession numbers of the sequences of the virus seeds used in MN assays should be included.

Response: As suggested, all the methods used were described in detail.

-          The reference reported should be carefully checked; reference no. 18 is reported in the main text but not in the reference list.

Response: References were checked and corrected

-          English language should be revised.

Response: English language has been revised, according to reviewers’ suggestions.

These are some minor comments:

Lines 31 and 34: the name in full of COVID-19 is reported twice

Response: The sentence has been corrected

  1. 41: When refer to VOC please add SARS-CoV-2 variant of concern

Response: The sentence has been corrected

  1. 41: Wu et al “have” demonstrated

Response: The sentence has been corrected

L.44: we “have” reported

Response: The sentence has been corrected

L.44: cell-mediated

Response: The sentence has been corrected

L.45: SARS-CoV-2, revise it throughout the manuscript

Response: The name has been corrected

L.54: add a reference to “our previous work”

Response: Reference was included

  1. 57: this sentence should be rephrased

Response: The sentence has been corrected

L.70: use the acronym for neutralizing antibodies (revise it throughout the manuscript)

Response: The sentence has been corrected

  1. 71: use “described” rather than “reported”

Response: The sentence has been corrected

  1. 77: clarify how results were expressed

Response: The sentence has been clarified

  1. 82: specify “after culture”

Response: The sentence has been corrected

  1. 95: delate the bracket after spots

Response: Bracket has been removed

L.100: add references to protocols used for molecular detection of SARS-CoV-2

Response: Method used for molecular detection has been specified

L.113. final dot is missing

Response: Final dot was included

  1. 115: “vaccinated” rather than “vaccinate” (check it throughout the manuscript)

Response: The sentence has been corrected

L.126: “at” rather than “At”

Response: The sentence has been corrected

  1. 137-138: this sentence should be clarified

Response: The sentence has been clarified

  1. 141-142: this sentence should be clarified

Response: The sentence has been corrected

  1. 152: the caption of figure 2 should be revised

Response: The caption of the figure has been changed

L.168: add the meaning of “NC” in the caption of figure 3

Response: Abbreviation was clarified

  1. 171: “subject” rather than “subjects”

Response: The sentence has been corrected

L-176: SARS rather than Sars

Response: The sentence has been corrected

L.180: this sentence should be clarified and report some quantitative data

Response: The sentence has been clarified and quantitative data have been included

L.182-185: this sentence should be moved to the discussion section. It is not relevant to the results section

Response: The sentence has been moved to discussion

L.190-193: this sentence should be moved to the discussion section. It is not relevant to the results section

Response: The sentence has been moved to discussion

L.203: “a negative value” rather than “a negative values”

Response: The sentence has been corrected

L.205: refer to “Alpha” and “Delta” VOCs

Response: Reference was included

L.209: “was similar” rather than “was not significantly different”

Response: The sentence has been corrected

L.216: “the results” rather than “the major results”

Response: The sentence has been corrected

L.220: in vitro re-stimulation

Response: The sentence has been corrected

L.226: “has been reported” rather than “have been reported”

Response: The sentence has been corrected

L.227: “studies” rather than “studies”

Response: The sentence has been corrected

L.252: Reference no 18 is missing

Response: References were checked and corrected

L.255: delete brackets for SARS

Response: Brackets have been removed

  1. 259: “limitations” rather than “limitation”

Response: The sentence has been corrected

L.260: do you mean elderly?

Response: Yes; the sentence has been corrected according to reviewer’s suggestion

L.275-277: missing

Response: The grant number has been included

Reviewer 2 Report

The authors presented humoral and cellular responses to anti-SARS-CoV-2 in SARS-CoV-2 naive individuals and nine previously infected individuals. Those patients were vaccinated with three mRNA vaccine doses. 

This work presents several flaws that limit the interest of this work.

* Which mRNA vaccine was used? It is already known that both mRNA vaccines most used until now present different responses after several months post-vaccination. Therefore, this information is pivotal to understanding this study. Moreover, a lot of hospitals use a heterogeneous vaccination for the third dose. It would be interesting to know the complete vaccination pattern. 

* The N for the previously infected patients is very low

* The results lack novelty, using the same cohort as in other works, the authors present another very similar study.

See "Evaluation of the Neutralizing Antibodies Response against 14 SARS-CoV-2 Variants in BNT162b2 Vaccinated Naive and COVID-19 Positive Healthcare Workers from a Northern Italian Hospital- Vaccines 2022 Apr29;10(5). doi: 10.3390/vaccines10050703". 

or

"Immunity to SARS-CoV-2 up to 15 months after infection- iScience. 2022 Feb 18;25(2):103743. doi: 10.1016/j.isci.2022.103743"

Moreover, apart from these authors, others showed similar results:

10.3390/vaccines10091563

10.3390/vaccines10091512

10.1038/s41598-022-19581-y  and so on.

* It's hard to define if there is a difference between T2 and T4 NT Abs, since the majority of results are superior to the detection limit. The authors should have diluted their samples to obtain this result. Since it is important to know if the boost induces a higher and stronger response or if the immune system is exhausted after three vaccine doses. 

*lines 153-154: not clear. would the authors say "at T3"? In general, several sentences are vague or not clear. The text must be verified.

*line 156. authors claim "but not at T4". This conclusion cannot be made because almost all the samples presented a BAU/ml >640. 

*One conclusion is that SARS-CoV-2 induced a higher level of CD4 response than CD8. This result was already published, for example, doi: 10.1080/21505594.2021.2019959.

The most interesting point is that the same result is found in previously infected individuals. The authors must have discussed this point. Why is there less response from CD8+T cells after the vaccine than CD4+?

* From 77 SARS-CoV-2 vaccinated individuals, 26 were infected. What are the variants? Moroever, the authors claim, "Levels of antibody and T-cell response after the booster dose was not significantly different in infected or non-infected subjects". Since the Abs titles were above the limit of detection, this conclusion cannot be made. 

* Are they statistical differences of the NT Abs and cellular response, between naive and experienced individuals at T2, T3 or T4?

* References are all mixed up. 

*Figure 1: the bars are not visible

*In Fig 3: What is the meaning of NC?

Author Response

The authors presented humoral and cellular responses to anti-SARS-CoV-2 in SARS-CoV-2 naive individuals and nine previously infected individuals. Those patients were vaccinated with three mRNA vaccine doses. 

This work presents several flaws that limit the interest of this work.

* Which mRNA vaccine was used? It is already known that both mRNA vaccines most used until now present different responses after several months post-vaccination. Therefore, this information is pivotal to understanding this study. Moreover, a lot of hospitals use a heterogeneous vaccination for the third dose. It would be interesting to know the complete vaccination pattern. 

Response: We included only subjects vaccinated with BNT162b2 mRNA vaccine

* The N for the previously infected patients is very low

Response: We totally agree with the reviewer about the low sample subset of SARS-CoV-2 experienced subjects and this point was included as study limitation at the end of discussion.

* The results lack novelty, using the same cohort as in other works, the authors present another very similar study.

See "Evaluation of the Neutralizing Antibodies Response against 14 SARS-CoV-2 Variants in BNT162b2 Vaccinated Naive and COVID-19 Positive Healthcare Workers from a Northern Italian Hospital- Vaccines 2022 Apr29;10(5). doi: 10.3390/vaccines10050703". 

or

"Immunity to SARS-CoV-2 up to 15 months after infection- iScience. 2022 Feb 18;25(2):103743. doi: 10.1016/j.isci.2022.103743"

Moreover, apart from these authors, others showed similar results:

10.3390/vaccines10091563

10.3390/vaccines10091512

10.1038/s41598-022-19581-y  and so on.

Response: We revised the discussion including the suggested references

* It's hard to define if there is a difference between T2 and T4 NT Abs, since the majority of results are superior to the detection limit. The authors should have diluted their samples to obtain this result. Since it is important to know if the boost induces a higher and stronger response or if the immune system is exhausted after three vaccine doses. 

Response: According with the reviewer suggestion, we know that a quantitative analysis is not possible since the majority of the results reached the detection limit of the assays. However, the rate of samples reaching the upper limit of assay detection is higher at T4 than respect to T2 and this result indirectly suggest the higher level of response at T4. To be more precise, we included the lack of quantitative analysis as study limitation.

*lines 153-154: not clear. would the authors say "at T3"? In general, several sentences are vague or not clear. The text must be verified.

Response: Correction was made

*line 156. authors claim "but not at T4". This conclusion cannot be made because almost all the samples presented a BAU/ml >640. 

Response: Correction was made

*One conclusion is that SARS-CoV-2 induced a higher level of CD4 response than CD8. This result was already published, for example, doi: 10.1080/21505594.2021.2019959.

The most interesting point is that the same result is found in previously infected individuals. The authors must have discussed this point. Why is there less response from CD8+T cells after the vaccine than CD4+?

Response: We remarked this point in discussion, according to the other results obtained

* From 77 SARS-CoV-2 vaccinated individuals, 26 were infected. What are the variants? Moroever, the authors claim, "Levels of antibody and T-cell response after the booster dose was not significantly different in infected or non-infected subjects". Since the Abs titles were above the limit of detection, this conclusion cannot be made. 

Response: We revised the conclusion, considering the reviewer’s suggestion.

* Are they statistical differences of the NT Abs and cellular response, between naive and experienced individuals at T2, T3 or T4?

Response: We observed differences between naïve and experienced subjects when T2 and T3 total IgG and NT Abs were analyzed but not at T4, probably because the large majority of the naïve subjects reached highest level of detectable response for both assays. Otherwise, looking at T-cell response, the only significant difference was observed at T3 between naïve and experienced subjects but not at T2 and T4. 

* References are all mixed up. 

Response: References were checked and corrected

*Figure 1: the bars are not visible

Response: Figures were modified

*In Fig 3: What is the meaning of NC?

Response: Abbreviation was explained in figure legend